# Meta-Heuristic Device-Free Localization Algorithm under Multiple Path Effect

**DOI:** 10.3390/e25071025

**Published:** 2023-07-05

**Authors:** Huajing Li, Ning Li, Yan Guo, Hao Yuan, Binghan Lei

**Affiliations:** College of Communications Engineering, PLA Army Engineering University, Nanjing 210007, China; lhj1121@foxmail.com (H.L.); yhlulita@163.com (H.Y.); leibinghan2000@163.com (B.L.)

**Keywords:** device-free localization multiple effects, compressed sensing theory, meta-heuristic algorithm

## Abstract

In the scenario of device-free localization under multiple effects, the accuracy of localization based on compressed sensing theory is severely affected. Most existing localization techniques directly ignore multiple path effects. However, it is not practical to ignore the multiple path effect due to its high signal strength, which can provide localization information. In this paper, we formulate the sensing matrix optimization problem in compressed sensing for device-free localization scenarios based on multiple reflections. To solve this problem, we model it as a constrained combinatorial optimization problem and propose a hybrid meta-heuristic algorithm. First, smart reflection surfaces and virtual node models are used to construct the desired communication links. Second, we iteratively improve the properties of the measurement matrix by using K-means clustering to obtain reasonable thresholds, and use a meta-heuristic algorithm to optimize the sensing matrix. Finally, the simulation results show that the proposed method efficiently optimizes the sensing matrix and achieves fast and high-precision localization while conserving communication resources.

## 1. Introduction

Individual positioning systems have become increasingly superior and are used in many fields such as intelligent transportation, national defense business, power dispatching, disaster relief and mitigation, and emergency search and rescue [1,2]. The more well-known BeiDou Navigation Satellite System and mobile cellular network-based localization system not only meet the current needs of human daily life, but also make full preparation for further reform of the localization system.

However, it is worth noting that the above two traditional positioning methods are no longer applicable when the target we want to locate is not able to carry a satellite signal receiving device, or when the target itself does not want to be monitored and located by satellite [3]. From the perspective of intelligent rail transportation, on the one hand, signal sensing and device-free positioning are the fundamental means to ensure the rapid development of intelligent transportation; on the other hand, from the perspective of security monitoring [4], it is a serious challenge for the intelligent rail transportation field to quickly determine emergency situations and respond to potential threats.

In addition, in the case of the GPS signal being relatively inferior and the satellite signal insufficient, the problem of the positioning and management of track line patrol personnel, as well as the safety monitoring of operators in the tunnel, should be given sufficient attention. Therefore, it is insufficient to rely solely on the locating system of the train itself to ensure the safe operation of urban tracks [5]. There should be additional device-free positioning technology to achieve all-round safety monitoring of the track, including the track line, stopping stations, and all objects in the tunnel. Take the train entry station in urban rail transit as an example, as shown in Figure 1.

When the train enters the station and stops, it is necessary to accurately locate each object in the station. In the face of such a complex scenario, academicians have proposed a device-free localization method that does not rely on the target to transmit wireless information.

Existing device-free locating technology uses two kinds of information: one is the received signal strength (RSS) [6], and the other is the channel state information (CSI) [7]. However, most wireless sensor devices can not directly use the fine-grained CSI. In this case, compared with the CSI, the information RSS carries by the wireless sensor is easy to obtain and the subsequent processing is relatively simple. Therefore, a device-free locating technique based on RSS is the preferred solution to the device-free localization (DFL) problem.

The efficient use of multiple path reflection signals in device-free localization is a highly trending research topic. The existing device-free localization algorithms to solve the multiple path reflection localization problem can be divided into the following four categories:(1)geometry-based device-free localization technology [8],(2)fingerprint-based device-free localization technology [9],(3)wireless laminar imaging-based device-free localization technology [10],(4)compressed sensing-based device-free localization technology [11].

Among them, the geometric method has low positioning precision, the fingerprint method is time consuming, and the wireless laminar imaging method has high hardware requirements. Moreover, compressed sensing technology emerges to provide a new idea to realize multi-target DFL. In the general localization scenario, the reflected signal, as the second intensities of the propagating signal, contains extremely abundant localization information, but it is not considered in the previous modeling and calculation of compressed sensing. In order to be able to effectively use multiple path signals, we construct a virtual node model in the compressed sensing theoretical system, use the received signal strength of the virtual sensor under the multiple path effect, and increase the communication without an additional hardware resource link.

Compressed sensing (CS) techniques enable the sensing of high-dimensional signals in uncorrelated observations in low-dimensional space through the effective use of signal sparsity. The target position vector is sparse in multi-target device-free localization; thus, only a small amount of measurement information needs to be collected through uncorrelated observations to reconstruct the target position vector with a very high probability. It is worth noting that the construction of the sensing matrix, an important step in compressed sensing, is inextricably linked to the accuracy of sparse recovery, and academicians have proposed many optimization methods to improve the sensing matrix. Donoho in [12] states that the sensing matrix should satisfy the condition that a certain degree of linear independence is maintained between all ranks. In CS theory, the general criteria used to evaluate perceptual matrices are finite isometry and mutual coherence, and in general, mutual coherence is easier to judge.

In order to ensure the linear independence between rows and columns as much as possible, scholars have proposed many optimization methods to improve the sensing matrix. However, the measurement matrix used in the device-free localization technology based on CS theory is a constrained binary sparse matrix determined by the sensor position [13], so in this case, it is impossible to optimize the sensing matrix. Therefore, most methods focus on improving the performance of the sensing matrix by multiplying the sensing matrix by a transformation matrix generated based on different principles. However, when considering noisy scenes, the method of multiplying by the change matrix will lead to a decrease in the signal-to-noise ratio, which affects the positioning accuracy. In the device-free localization scenario with virtual sensor nodes, the design of the sensing matrix includes the signals of both real sensors and virtual sensors. At the same time, the signal propagation model is used in the localization scene, so it is difficult to simply optimize the sensing matrix because the virtual sensors are involved in the construction and the geometric topology formed by the grid points of the localization area division.

Therefore, this paper proposes a hybrid meta-heuristic algorithm to optimize the sensing matrix, which optimizes the measurement matrix composed of a constrained binary sparse matrix through an iterative method, and establishes the model of combinatorial optimization. At the same time, in order to better describe the performance of the optimized sensing matrix, the mutual coherence of t% described in [14] is used as the description feature. Here, t refers to the concept of mutual coherence proposed by reference [14], which is explained as follows. The mutual coherence provides a measure of the worst similarity between the dictionary columns, a value that exposes the dictionary’s vulnerability; as such, two closely related columns may confuse any pursuit technique. In order to deduce the threshold t, the optimal threshold is selected using the clustering method. In addition, the hybrid meta-heuristic algorithm is used to solve the combinatorial optimization problem established above so as to realize the optimization of the sensing matrix.

The main contributions made in this paper are as follows:(a)In order to improve the performance of the device-free localization algorithm under multiple path effects, the optimization problem of the sensing matrix based on compressive sensing theory is studied. In the device-free localization scenario constructed with real and virtual sensors, the measurement matrix is still guaranteed to optimize the sensing matrix iteratively under the constraint of a binary sparse matrix, and the signal-to-noise ratio of the RSSI is guaranteed not to decrease.(b)The characteristics of the sensing matrix are analyzed based on virtual sensors and real sensors in the device-free localization scene under the multiple path effect, and the t% is used as the evaluation criterion of the mutual coherence feature. To judge the merits of each measurement matrix and determine the optimal threshold *t*, the diagonal elements of the Gram matrix are classified using a clustering algorithm, and the optimal threshold is obtained based on the results of different clusters. Based on the obtained optimal index, M-HTLS (Meta-Heuristic Tabu Local Search) is proposed to solve the constructed combinatorial optimization problem.(c)A large number of simulation results show that CS localization methods using different recovery algorithms can optimize the sensing matrix using M-HTLS, and under the same conditions, compared with traditional optimization methods, the localization error is much smaller.

The remaining part of the paper is organized as follows: Section 2 presents the existing methods for sensing matrix optimization and existing technological advances in device-free localization. In Section 3, a device-free localization system model based on the virtual sensor model is constructed from the solution of the inverse ray-tracing algorithm and a combinatorial optimization problem for the sensing matrix is formulated. Section 4 leads to the M-HTLS-based sensing matrix optimization method. In Section 5, the superiority of the proposed method is validated by extensive simulation results in MATLAB and specific correlation performance analysis. Section 6 concludes the paper with a vision for future work.

## 2. Related Work

Many experts have put forward the improvement or design method of the sensing matrix, because the sensing matrix is obtained by the product of the measurement matrix and the sparse dictionary, and the sparse dictionary is determined by the sensing signal to be recovered; thus, experts are focused on the design of the measurement matrix to optimize the sensing matrix. The design of the measurement matrix can be generally divided into two approaches. The first approach is to find a suitable random measurement matrix such as a Gaussian, Hadamard, or Bernoulli matrix as the measurement matrix. The elements of matrices of this type are distributed independently and randomly, and require considerable storage space. Due to its intrinsically unstructured nature, the computation is complicated and the hardware requirements are too high for practical applications. The second approach, in contrast to random matrices, involves deterministic matrices, where the elements of the matrix are designed by the researcher and are therefore deterministic. In recent years, many approaches have been proposed by academicians on how to design deterministic measurement matrices.

In [15], specific codes and optimal codebooks are used to generate a deterministic CS matrix. In [16], the determined measurement matrix is constructed through Bose balanced incomplete blocks, while improving the flexibility of the embedding operation. Reference [17] applies a sparse fast Fourier transform in the determined measurement matrix. In addition, ref. [18] also propose a deterministic construction method of bipolar measurement matrix based on the binary sequence family. At the same time, many works optimize the sensing matrix by improving its performance. In [19], singular value decomposition is used to improve the sensing matrix, which ensures the sparsity of the original signal and the restricted isometry property (RIP) of the new sensing matrix, but the computational cost is large. The researchers in [20] optimize the sensing matrix by multiplying the orthogonal form of the sensing matrix with its pseudo-inverse. The optimization method adopted in [21] is based on equiangular noncoherent cell norm compact frame sensing matrices, and a random matrix is used as the initial preconditioning matrix. An optimal Frobinius norm compact frame is obtained as the optimal sensing matrix by iteratively relaxing the Gram matrix and employing matrix algebra decomposition. Essentially, they optimize by multiplying the sensing matrix by a transformation matrix generated based on different principles. However, the effectiveness of these methods may decrease when measurement noise is taken into account, as the SNR decreases when multiplied with the transformation matrix, resulting in a higher probability of misestimation.

## 3. System Model and Problem Formulation

In this paper, the virtual sensor node model is designed using the localization signal reflected from the intelligent reflective surface mentioned in [22], as shown in Figure 2. We can take advantage of the intelligent reflective surface that can independently control the incident signal amplitude to reduce the loss of the RSS value and provide a guarantee for the accuracy of localization. With the virtual sensor node model, the reflected signals of a large number of multiple path components are modeled as signals from a mirror sensor behind the reflecting surface, such that the physical and virtual sensors are placed inside the rectangle formed by the intelligent reflecting surface. The deployment pattern of the alternating arrangements of real and virtual sensors can be specified as follows:(1)$$r_{v}(rx)=\frac{\left[r_{a}(rx)-s_{k}\right]}{P_{k}}$$
where $P_{k}$ is the symmetric orthogonal matrix, which can be expressed as $P_{k}=E-2W_{n}(kx)W_{n}{\left(kx\right)}^{T}$, and $s_{k}$ denotes the offset, which can be expressed as $s_{k}=2∥r_{v}(rx)-k_{x}∥$. $r_{t}$ represents the real sensor in the virtual sensor node model. $r_{a}$ indicates the mirror sensor of the real sensor to the IRS. However, $W_{n}(kx),k=1,2,3,4$ is the normal vector of the reflecting surface IRS, which can be represented by homogeneous coordinates $\left(A_{x},B_{x}\right)$, so the normal vector can be expressed as follows:(2)$$W_{n}(kx)={\left(A_{x}^{2}+B_{x}^{2}\right)}^{-\frac{1}{2}}{\left(A_{x},B_{x}\right)}^{T}$$

According to the solution of the reverse ray-tracing algorithm [23], if the original total number of links is L, through the virtual sensor model, L + V links will be obtained, where V is the link obtained from the virtual sensor obtained by reverse ray tracing. It can be seen that the communication links are increased without increasing the hardware consumption.

### 3.1. System Model

There are K targets randomly distributed in a designed rectangular localization region consisting of intelligent reflecting surfaces. To estimate their positions, sensor nodes capable of collecting relevant RSS information should be deployed around a rectangular localization region consisting of smart reflecting surfaces. Figure 3 shows how this structure is deployed. In this paper, we construct the signal reflected by an intelligent reflecting surface as the signal emitted from a mirror sensor behind the intelligent reflecting surface. A virtual sensor node is called when the signals sent by several mirror sensors converge at a position between two real sensors. The received signal strength information of the real and virtual sensors acting on the communication link is used to calculate the difference of the received signal strengths using this localization information. In order to better cover the positioning area, it is assumed that the number of these real sensors and virtual sensors deployed uniformly on the boundary of the positioning area is 2M, and every two sensors (one sending and one receiving) form a communication link, so there are M communication links. We store the initial received signal strength value of M communication links (the signal strength value when there is no occlusion is taken as the initial value) in the vector $y_{0}$, and then the difference in the received signal strength y can be calculated as:(3)$$y=y_{k}-y_{0}$$
where $y_{k}$ is the received signal strength value when there is a target to be located in M communication links with occlusion.

### 3.2. Problem Formulation

Since the theory of compressed sensing is aimed at discrete signals, we need to mesh the location region in order to apply the sparse recovery algorithm in compressed sensing to device-free locations. The localization area is divided into *N* grids of the same size and numbered in order, namely, 1,2,3,…,n,…,N. An *n*-dimensional vector $\theta_{N\times 1}$ is used to represent the position distribution of *K* targets (K<<N) to be located: if a grid contains targets, the coordinate position of the center point of the grid in the rectangular coordinate system is regarded as the target position, the value of the corresponding position in the $\theta_{N\times 1}$ vector is set to 1, and the other positions without targets are set to 0. The localization scenario is shown in Figure 3. At this time, using the knowledge of compressed sensing theory, the signal strength difference vector y can be expressed as follows:(4)y=Φθ+ε
where ε is the noise vector and θ is an *N*-dimensional vector. The vector θ is sparse due to the number of non-zero elements K≪N. Therefore, we transform the position estimation problem into a sparse signal recovery problem, which is to recover a sparse signal θ by solving a norm $l_{0}$ minimization problem with a known sensing matrix Φ and vector θ, as follows:(5)$$\tilde{\theta }=arg\;min{\left\|\theta \right\|}_{0}\begin{array}{ccc} & s.t. & y=\Phi \theta +\varepsilon \end{array}$$

As long as the values of the nonzero elements of θ are estimated, the target position ${\left\{p_{K}\right\}}_{K=1}^{K}$ is determined.

There are many sparse recovery algorithms based on compressive sensing theory, such as Basis Pursuit (BP) [24], Orthogonal Matching Pursuit (OMP) [25] and Sparse Bayesian Learning (SBL) [26]. However, a high degree of coherence between the columns in Φ would severely confuse any recovery algorithm and lead to poor localization results. We note that in the multi-target localization scenario based on multi-path effect CS, the columns of the sensing matrix are highly correlated due to the deployment of virtual and real sensors and the distribution of target locations to be located. Therefore, the sensing matrix used in this scenario should be optimized to achieve higher localization accuracy.

In the system of compressed sensing theory, the sensing matrix Φ can be expressed as:(6)Φ=HΨ

Among them, H is a 0–1 sparse measurement matrix determined by the sensor deployment. Every row of H has only one non-zero element and every column has, at most, one non-zero element. Ψ is a sparse dictionary whose elements are in the i-th row and the j-th column can be represented by $\varsigma (\hbar_{i},{ƛ}_{j})$. Technically, the product of matrix H as well as Ψ can be regarded as the product by M×∞ and ∞×N. Element $H_{i,j}$ in H represents the locations of real and virtual sensors. Consequently, (6) indicates that a subset Φ can be formed by selecting row M from Ψ. For this purpose, the number of virtual sensor locations is limited to a finite value N and they are ordered like the grid shown in Figure 3. The size of the matrix H and the matrix Ψ can be reduced to M×N and N×N. Hence, the optimal distribution of the non-zero elements of the measurement matrix should be found to minimize the column similarity in Φ.

## 4. Sensing Matrix Optimization Method Based on M-HTLS

The stated problem in the previous section is a combinatorial optimization problem. In this section, we propose a hybrid meta-heuristic algorithm called M-HTLS to solve this problem.

First, we use t% average mutual coherence as an indication to evaluate the optimality of different sensing matrices. Then, we use K-means mean clustering to classify the non-diagonal elements of the Gram matrices to derive a threshold t. Next, M-HTLS is used to solve the combinatorial optimization problem based on this threshold to optimize the perceptual matrices. In conclusion, we can improve the accuracy of multi-target localization based on multiple path effect CS with the optimized sensing matrix. Figure 4 shows the flow chart of the proposed sensing matrix optimization method in this paper.

### 4.1. Quantitative Index

In order to quantify the “qualitative” of the different sensing matrices and thus help in the recovery of sparse signals, it is necessary to find a suitable metric, a well-known metric of mutual coherence μ. Then, for the sensing matrix Φ, the similarity of its column i and column j can be expressed as follows:(7)$$m_{ij}=\frac{\left|d_{i}^{T}d_{j}\right|}{\left\|d_{i}\right\|\cdot \left\|d_{j}\right\|}$$

Then, the mutual coherence of the sensing matrix can be expressed as:(8)$$\mu \left\{\Phi \right\}=\underset{1\leq i,j\leq N,i\neq j}{\max}m_{ij}$$

Equation (8) represents the largest normalized inner product among all inner products between any two different columns in Φ. With metric μ, we can quantify the matrix with the strongest column correlation, because in the worst case, metric μ has the strongest similarity when it contains θ columns. The matrix with the strongest column correlation is severely confounded in the recovery of sparse signals when the CS recovery algorithm is used. In this case, a larger μ{Φ} leads to worse sparse recovery results.

The Gram matrix also has the concept of mutual coherence, which can be expressed as:(9)$$G={\tilde{\Phi }}^{T}\tilde{\Phi }$$
where $\tilde{\Phi }$ is called the effective dictionary and is obtained by normalizing the inner dictionary product of all the column elements of Φ and taking their absolute values. The non-diagonal element $g_{ij}$ in G represents the normalized inner product between the i-th and j-th columns of Φ and taking its absolute value, the maximum value is exactly equal to μ{Φ}.

From the literature [27], we know that sparse recovery algorithms such as BP and OMP are able to find an exact solution to Equation (5) if the following inequalities are satisfied:(10)$${\left\|\theta \right\|}_{0}<\frac{1}{2}\left(1+\frac{1}{\mu \left\{\Phi \right\}}\right)$$

Since the metric of mutual coherence in Φ is characterized by the maximum column similarity above, which reflects the recovery of sparse signals under the most extreme conditions, it may not be a reasonable measure of the actual “quality” of Φ. If we focus not only on the maximum value, but also on the average value, then it can more reasonably reflect the actual performance of sparse recovery of Φ. Therefore, the focus should be on the average mutual coherence t%, which is defined as follows:(11)$$\mu_{t\%}\left\{\Phi \right\}=\frac{\underset{1\leq i,j\leq N,i\neq j}{\Sigma }\left(g_{ij}\in G_{t}\right)\cdot \left|g_{ij}\right|}{{\left\|G_{t}\right\|}_{0}}$$
where $G_{t}$ is the set of off-diagonal entries of G. The above equation filters out relatively small elements of G, and only targets larger elements. $\mu_{t\%}\left\{\Phi \right\}$ has a different meaning for different thresholds, and for t=100, denotes the average of all off-diagonal elements of G. As t decreases, $\mu_{t\%}$ gradually rises, and for $t=\frac{100}{{\left\|G\right\|}_{0}-N}$, $\mu_{t\%}=\mu$. In the following sections, we discuss the selection method for the threshold t and the combinatorial optimization problem below, and adopt the average mutual coherence as the quantitative metric to optimize the sensing matrix.

The first is the method of choosing the threshold t. In the technique of CS-based device-free localization with a multiple path effect, we find that the column similarity features of Φ are as follows: assuming that $d_{i}$ is the i-th column in Φ, the closer the columns are to each other, the higher the similarity, while for those columns that are far apart, the similarity is low, where the “distance” between columns is calculated according to the following equation, such as the “distance” between i and j:(12)$$I_{ij}={\left\|\ell_{i}-\ell_{j}\right\|}_{2}$$
where $\ell_{i}$ and $\ell_{j}$ both represent the two-dimensional coordinates analytically. Taking $\ell_{i}$ as an example, its two-dimensional coordinates can be expressed as follows:(13)$$\begin{array}{l} x_{\ell_{i}}=\{\begin{array}{cc} \mathrm{mod}\left(i,n\right), & \mathrm{mod}\left(i,n\right)\neq 0\\ n, & otherwise \end{array}\\ y_{\ell_{i}}=\left\lceil \frac{i}{n}\right\rceil \end{array}$$

For clearer identification, a color depth map is used to represent the similarity between column i-th, $d_{i}$, and the other columns in Φ. As shown in Figure 5, which represents the similarity between column 58-th, $d_{58}$, and other columns in Φ, we can roughly divide the colors into three categories: the “high value area” is represented in yellow, the “low value area” is represented in blue, and the rest is called the “middle area”. Thus, we can also divide the non-diagonal elements in G into three categories. Among them, the category with the highest mean value determines the range to be optimized. For this purpose, we use the K-means clustering algorithm, which is able to obtain optimal clustering results quickly for one-dimensional data, to classify all non-diagonal elements in G and determine the threshold value t.

In order to reflect the real characteristics of the localization scene, Φ is obtained in a random way, where n=10, i= 58, M=30, N=100 and the target to be located is randomly distributed in the area, and the real and virtual sensors are arranged at the boundary. As shown in Figure 5, lighter colors indicate a stronger similarity between columns, and vice versa. The grid of the circle stay is denoted as $d_{58}$, and for the other columns of the grid, the closer the circle is, the stronger the similarity. For example, the coordinates of triangles and squares are [4,7] and [7,5], which represent $d_{47}$ and $d_{75}$, respectively, according to the equation above. The color around the triangles is lighter than the color surrounding the squares, which means a stronger similarity due to the shorter distance.

In overview, we can judge that the larger the value of certain non-diagonal elements in Φ, the more thoroughly the performance of the CS method deteriorates, so we need to improve all non-diagonal elements in G in a targeted way, and select them by determining the threshold t. We divide them into W classes, $S_{1},S_{2},\cdots ,S_{w}$, respectively, and each class has a focal point $o_{w}$. We use the Euclidean distance as the criterion for assessing the similarity between the sample points. $o_{w}$ should be equal to the average of all samples in $S_{w}$ to minimize the sum of quadratic distances between the focal point and all points in $S_{w}$, using the least squares method and Lagrange’s principle, which can be expressed as:(14)$$J\left(S_{w}\right)=\underset{g_{ij\in S_{w}},i\neq j}{\Sigma }{\left\|g_{ij}-o_{w}\right\|}^{2}$$
where we define the criteria for clustering functions as:(15)$$J=\overset{W}{\underset{w=1}{\Sigma }}\underset{g_{ij\in S_{w}},i\neq j}{\Sigma }{\left\|g_{ij}-o_{w}\right\|}^{2}$$

Based on the prior analysis, we set W=3. First, the W elements are selected from all non-diagonal terms of G as the initial focus. Second, the remaining samples are assigned to the categories holding different focuses according to the minimum distance criterion, and then W clusters are obtained. Next, iteration *i* updates the cluster mean of each focus and continuously recalculates the clustering criterion function J. Until the focus and J remain unchanged, we consider the separation of all sample points complete. Otherwise, we continue with the second step and iterate to perform the above process. Finally, we can obtain:(16)$$t=\frac{{\left\|S^{\alpha }\right\|}_{0}}{{\left\|G\right\|}_{0}-N}\times 100$$
where $S^{\alpha }$ is the categorical set with the highest mean column correlation.

### 4.2. Sensing Matrix Optimization Using M-HTLS

As previously mentioned, in order to obtain more accurate localization results, we need to optimize the sensing matrix to find the minimum value of $\mu_{\%}\left\{\Phi \right\}$. Considering the constraints on Φ, we can transform the perceptual matrix optimization problem into an NP-hard constraint combination optimization problem, as follows:(17)$$\begin{array}{l} \arg_{H}\min\mu_{t\%}\left\{H\Psi \right\}\\ s.t.\\ \begin{array}{ccc} & \overset{N}{\underset{j=1}{\Sigma }} & h_{ij}=1,i=1,2,\cdots ,M \end{array}\\ \begin{array}{ccc} & \overset{M}{\underset{i=1}{\Sigma }} & h_{ij}\in \left\{0,1\right\} \end{array},j=1,2,\cdots ,N \end{array}$$

The algorithms used to solve this problem can be divided into exact and approximation algorithms. Although the exact algorithm can give a globally optimal solution, this comes at the expense of computational time in the search space. However, approximate algorithms can give near-optimal solutions in reasonable computational time, and can be further divided into heuristic and meta-heuristic algorithms. Heuristic algorithms, such as Local Search (LS), are effective in exploring the neighboring solution space, but are prone to be locally optimal. Meta-heuristic algorithms can remove local optima and extensively search for global optima, such as Genetic Algorithm (GA), which combines a better solution with a higher fitness to obtain an approximate optimal solution. The genetic idea was originally proposed by Holland under the inspiration of “survival of the fittest” [28]. At each iteration, individuals with higher fitness have a greater chance of survival than those with lower fitness, and the loop iteratively screens out individuals with higher fitness to obtain the final ideal solution. However, GA is not good at finding the optimal solution deep in a certain region, so we optimize it with a hybrid strategy.

Adopting the idea of hybrid algorithms, where heuristics and genetic algorithms complement each other, has proven to be a good attempt in recent research. We propose a hybrid meta-heuristic, M-HTLS, in which the tabu strategy is combined with local search to enhance the local search capability of GA. Compared with the original LS, Tabu Local Search (TLS) can shorten the computation time in the optimization process, so as to save the computation cost.

M-HTLS starts with a randomly generated population, where each individual represents a measurement matrix. Since the size of the measurement matrix H is quite large, for simplicity, the “gene sequence” x∈ℝM saved in each individual is used to represent each measurement matrix. Each gene sequence contains M distinct integers ranging from 1 to N. If xm=n, then hmn=1, otherwise hmn=0. In this way, the various measurement matrices are compressed into different gene sequences, while the equality constraints can also be satisfied. Assuming a population size of Np, its gene pool can be represented by X={xi}i=1Np.

Next, a loop consisting of a fitness function, parent selection, TLS, crossover, and mutation is iteratively performed to update the population until the end criterion is satisfied. The top F% individuals are selected based on their fitness and TLS is performed on them. By “top,” we mean that individuals with higher fitness have a better chance of surviving, so TLS makes sense. In general, the larger the fitness function, the better the optimization, but, obviously, the higher the computational cost. In order to achieve equilibrium, F=10 is set.

From the above, the LS algorithm needs to iteratively search the neighborhood space for a locally optimal solution until the neighborhood candidates are exhausted. Although the results must be optimal, the computational cost is also quite expensive; in particular, the calculation of the fitness function is more complicated, by:(18)$$F(x)=1-\mu_{t\%}\left\{\Phi \right\}=1-\mu_{t\%}\left\{H\Psi \right\}$$

Above all, an M-HTLS algorithm is proposed by analyzing the correlation properties of the fitness function. In this algorithm, the TLS algorithm can achieve a higher time efficiency than the original LS algorithm in the optimization process.

The non-diagonal entries $g_{m,n}(m\neq n)$ of Gram matrix G are given by Equation (19). $x_{j}$ determines the position of the nonzero element in row j of H, which determines that the row in Ψ will be affected.
(19)$$\begin{array}{l} g_{m,n}=\frac{\overset{M}{\underset{i=1,i\neq j}{\Sigma }}\psi_{x_{i},m}\psi_{x_{i},n}+\psi_{x_{j},m}\psi_{x_{j},n}}{\sqrt{\overset{M}{\underset{i=1,i\neq j}{\Sigma }}\psi^{2}{}_{x_{i},m}\psi^{2}{}_{x_{j},m}}\sqrt{\overset{M}{\underset{i=1,i\neq j}{\Sigma }}\psi^{2}{}_{x_{i},n}\psi^{2}{}_{x_{j},n}}} \end{array}$$

According to Cauchy’s inequality, $g_{m,n}$ takes its maximum when $\frac{\psi_{x_{j},m}}{\psi_{x_{j},n}}=\frac{\psi_{x_{1},m}}{\psi_{x_{1},n}}=\cdots =\frac{\psi_{x_{M},m}}{\psi_{x_{M},n}}$, but experiments show that the Gram matrix G is not satisfied when $\frac{\psi_{x_{1},m}}{\psi_{x_{1},n}}=\cdots =\frac{\psi_{x_{M},m}}{\psi_{x_{M},n}}$, so the maximum of $g_{m,n}$ should fall in the range of (min{R},max{R}), $R=\left\{\frac{\psi_{x_{i},m}}{\psi_{x_{i},n}},i=1,\cdots ,j-1,j+1,\cdots M\right\}$. When the ratio $\frac{\psi_{x_{j},m}}{\psi_{x_{j},n}}$ deviates from this range, the value of $g_{m,n}$ becomes smaller. From this, it can be concluded that the larger the difference between $\psi_{x_{j}}$ and the remaining M−1 columns, the smaller the sum of the off-diagonal entries of G, thus reducing $\mu_{t\%}\left\{H\Psi \right\}$ more and improving the fitness F(x).

Based on the above conclusions, the tabu list is generated according to the TLS step so that invalid solutions can be filtered out from the solution neighborhood. In the multiple reflection device-free localization scenario, the difference between $\psi_{x_{i}}$ and $\psi_{x_{j}}$ increases as the distance between the position of the $x_{i}$ sensor and the $x_{j}$ sensor is extended, and vice versa. With the deployment of the preset real and virtual sensor locations, the closest to the preset sensor location is $x_{i-1},x_{i+1},x_{i-n},x_{i+n}$, respectively, so that the tabu table can be listed as $T=\left\{x_{i}-1,x_{i}+1,x_{i}-n,x_{i}+n,1\leq i\leq M\right\}$. The motifs are selected from the population by the fitness of the individuals, which is determined by the fitness function F(x). Specifically, the survival probability is proportional to the value of F(x). As a result, individuals with high survival rates are more likely to be selected as parents. Here, the threshold t is obtained by the above K-means clustering algorithm. To make the threshold more representative, $t_{1},t_{2},\cdots ,t_{N_{p}}$ was computed from $\Phi_{1},\Phi_{2},\cdots ,\Phi_{N_{p}}$ and averaged to obtain the final threshold t.

After selection, the parents are paired randomly with a reproduction probability $P_{c}$. In this process, the parental gene sequences are crossed using a multi-point mechanism. In a word, this kind of rule is to randomly decide a few gene loci; that is, the selected gene does not contain the gene sequence of the other side, so as to satisfy the constraint in Equation (17). Parents then exchange genes at these loci to produce two offspring, and for those parents who do not pair, the offspring copy their genes directly. The crossover process enables the characteristics of the parent generation to be passed on to the offspring.

Parent selection and crossover operators pick out good individuals according to the “survival of the fittest” principle, but this may reduce the diversity of the population, because most populations will share a similar gene sequence after only a few generations. This premature convergence will prevent the algorithm from finding the optimal solution. Thus, a mutation operator is added after the crossover operation to determine the individual mutation with probability $P_{m}$, a small value, here set to 0.01. Individual mutation is carried out by randomly selecting a gene locus on its sequence, replacing the original integer with a different integer from 1 to *n*, and setting the end criterion as the evolutionary iteration reaches a threshold $e_{\max}$.

In the iterative loop above, the quasi-optimal solution $x^{*}$ can be obtained from the updated population. Until the final stopping criterion is satisfied, the optimized sensing matrix $\Phi^{*}$ can be obtained. The parameters in $N_{p}$, $P_{c}$, and $P_{m}$ should be assigned in advance in the GA. Algorithm 1 shows the flow of the proposed sensing matrix optimization method.
**Algorithm 1.** Sensing matrix optimization method**Initialize**:1. Set $N_{p}=100$, Q=10, $P_{c}\;=\;0.8$, $P_{m}=0.01$, $e_{\max}=300$, e=0;2. The initial population $X={\left\{x_{i}\right\}}_{i=1}^{N_{p}}$ is generated randomly;3. Solve the calculation $\Phi_{1},\Phi_{2},\ldots ,\Phi_{N_{p}}$;4. Calculate $t_{1},t_{2},\ldots ,t_{N_{p}}$, and the threshold t is obtained;5. Repeat;6. Compute the fitness F(x) of the individuals of the population;7. The top Q% individuals are selected according to their fitness size for TLS operation;8. The parent selection operation is performed according to the individual fitness;9. The parents are paired with the crossover probability $P_{c}$, and the multi-point crossover operation is performed;10. Variant operation is performed with variant probability $P_{m}$;11. Update the population X, e←e+1;12. Until $e\geq e_{\max}$;13. The individual $x^{*}$ with the highest fitness is selected from X and the optimization measurement matrix $H^{*}$ is solved;14. Optimize the sensing matrix $\Phi^{*}=H^{*}\Psi$.

In the proposed sensing matrix optimization algorithm, the main computational time cost is used in the optimization process of K-means clustering and hybrid meta-heuristics. For the K-means procedure, the computational complexity is $O\left(WN^{2}T\right)$, where T is the iteration time and $N^{2}$ is the number of classes of categorical data, here W=3. Since W is constant and $T\ll N^{2}$, its complexity can be written as $O\left(N^{2}\right)$. For the GA process, the computational complexity is $O\left(N^{2}\log_{2}N\right)$ since the population size and the number of iterations remain the same. The complexity depends mainly on the fitness function, and the computational cost is mainly reflected in the heap sorting process. The computational complexity of LS is $O\left(N^{3}\log_{2}N\right)$. In the TLS procedure, on average N−χM neighboring solutions are omitted compared to LS; hence, the computational complexity of M-HTLS is $O\left(\left(N-\chi M\right)N^{2}\log_{2}N\right)$, where χ is a constant and the range of values is χ∈(1,5).

## 5. Numerical Results

In this section, the effectiveness and robustness of the M-HTLS algorithm for CS sensing matrix optimization in the multi-path reflection device-free localization scenario will be elaborated through extensive simulation experiments.

### 5.1. Simulation Setup

The simulation platform used in this paper is MATLAB 2018b. The relevant parameter settings in the algorithm are listed in Algorithm 1. By default, we set the target area as a 10 m × 10 m area and divide it into N=100 grid areas with side lengths of 1 m, so the number of grids on each side is n=10. The K=4 localization targets are randomly distributed within the localization scene region, and M=30 real and virtual sensor locations are selected from S=100 uniformly distributed candidate link locations determined by the measurement matrix used. In addition, the noise added to the RSS measurements is modeled as a vector ε following a zero-mean Gaussian distribution $N\left(0,\sigma^{2}\right)$, where each term in ε is kept independent. As for the calculation of SNR, we refer to reference [29,30], so in the simulations, the SNR is calculated using $10\mathrm{lg}\left({\left\|\Phi \theta \right\|}_{2}^{2}/M\sigma^{2}\right)$ with a default initial value of 10 dB. In order to effectively compare optimization methods, we define the average localization error as the evaluation metric, as follows:(20)$$Ave.Err=\frac{\overset{K}{\underset{k=1}{\Sigma }}{\left\|t_{k}-{\tilde{t}}_{k}\right\|}_{2}}{K}$$
where $t_{k}$ and ${\tilde{t}}_{k}$ are the true and estimated positions of the k-th targets, respectively. When computing the Ave.Err, in order to ensure the fairness and rationality of the experiment, we will find the one-to-one correspondence between the true position and the estimated position of the target to be located to minimize error. All numerical results shown in the performance comparison section for the various algorithms of the simulations are averaged over 500 Monte Carlo simulations.

### 5.2. Performance Comparison among Different Sensing Matrix Optimization Methods

By the above derivation, it follows that the value of the threshold t is 9.872; hence, the fitness function in GA can be expressed as $1-\mu_{9.872\%}\left\{\Phi \right\}$.

The simulation results in Figure 6 show the performance comparison between M-HTLS, GA, and M-HLS. Each curve is obtained by averaging the results of 20 experiments, and the average time over 200 iterations is presented for all algorithms. As the number of evolving generations increases, the value of $\mu_{t\%}\left\{\Phi \right\}$ decreases continuously as the three algorithms are optimized until the end of the iteration. As shown in Figure 6, GA converges much slower than M-HLS and M-HTLS. However, M-HLS and M-HTLS achieve better optimization results at the expense of computational time. Compared to the GA algorithm, M-HLS improves the performance by 0.023 when the additional time cost increases by 235.2 s. However, the proposed M-HTLS can improve the optimization results by 0.015 at an additional time cost of 42.2 s. Considering the trade-off between convergence speed and time cost, we use M-HTLS as the sensing matrix optimization algorithm and evaluate its performance in the following aspects.

As shown in Figure 7, the optimization effect of the proposed M-HTLS on the sensing matrix can be seen. The green and red bars represent the distribution before and after optimizing the off-diagonal entries of G, respectively. It is clear that before the optimization, the off-diagonal terms occupy a high fraction of the population, while after the optimization, most of the off-diagonal terms in G are optimized to the low-value region. Thus, it is known that the similarity of the sensing matrix Φ is reduced and, hence, the localization accuracy is improved.

The results are shown in Figure 8, which clearly shows the optimization effect of the proposed M-HTLS on three CS theory-based sparse recovery methods, namely, BP, OMP, and SBL, in different SNR environments. It can be seen that the error rates of the three recovery methods decrease as the SNR value increases from 10 to 40, which is in agreement with the actual situation. In the initial case, when the SNR value is 10, the average error values of the three sparse recovery algorithms are higher for the random sensing matrix than for the optimized sensing matrix. Using OMP as an example, the average error value before optimization is 2.09, while the average error value after optimization is drastically reduced to only 1.16, resulting in a 44.7% reduction in the error rate. When the SNR value is increased to 40, the channel environment is theoretically more ideal and the average error of the unoptimized sensing matrix is still higher than that of the optimized sensing matrix. Using BP as an example, the average error before optimization is 0.62 and the average error after optimization is 0.12, resulting in an error rate reduction of 80.7 percent. It can be seen that the proposed algorithm can show excellent performance for different SNRs and various sparse recovery methods

This is shown in Figure 9, which clearly shows the optimization effect of the proposed M-HTLS on the three sparse recovery methods based on CS theory, namely, BP, OMP, and SBL, for different numbers of targets to be located. It can be seen that the average localization error value rises when the number of targets to be located increases from 1 to 7, which is in line with reality. At the beginning of the iteration, when the number of targets to be located is one, the localization difficulty is low and the localization error of each recovery method is small. Using the OMP method as an example, when the target to be located is 1, the average localization error without optimization is 0.77 and after optimization is 0.55, resulting in a 28.2% reduction in the error rate. When the number of targets to be located is seven, the average localization error without optimization is 1.43 and after optimization is 1.08, resulting in a 24.7% reduction in the error rate. The BP and SBL methods can also reduce the average error, which can show the superiority of the proposed optimization algorithm.

As shown in Figure 10, the optimization effect of the proposed M-HTLS on the three CS theory-based sparse recovery methods, BP, OMP, and SBL, with different numbers of sensing matrices is represented. It is common knowledge that the larger the number of optimized sensing matrices, the smaller the average localization error value should be, leading to improved localization accuracy. The simulation results show that this is in agreement with the expected situation. In the initial case, where the number of sensing matrices is set to 15, the mean localization error values of the three sparse recovery methods after optimization are smaller than those before optimization. When the number of optimized sensing matrices reaches 45, the average localization error is greatly reduced and the average error value of each method after optimization is smaller than before optimization. Using OMP as an example, when the number of optimized sensing matrices is 15 in the initial stage, the average error of the non-optimized case is 1.93, while the average error value of the optimized case is reduced to 1.51 and the error rate is reduced by 21.7 percent. When the number of optimized sensing matrices is increased to 45, the average error of the unoptimized sensing matrix is 0.88 and the average error of the optimized sensing matrix is 0.70, resulting in a 20.7% reduction in the error rate. Similarly, the BP and SBL methods can also reduce the average error as the number of optimized sensing matrices increases. It can be seen that the proposed method can reduce the average error value and improve the localization accuracy.

## 6. Conclusions and Future Work

In this paper, we investigate the CS sensing matrix optimization problem in a device-free localization scenario based on multiple reflections. To solve this problem, we model it as a constrained combinatorial optimization problem and propose a hybrid meta-heuristic algorithm M-HTLS. In doing so, we iteratively improve the properties of the measurement matrix while keeping it as a constrained binary sparse matrix. The average mutual coherence of t% is used as an optimization metric to evaluate the quality of each sensing matrix. Moreover, the K-means clustering algorithm determines an appropriate threshold t, based on the characteristics of the sensing matrix in this scenario. In this paper, we show that M-HTLS avoids generating additional transformation matrices, which does not lead to the degradation of the signal-to-noise ratio compared to the conventional CS-based localization methods. Therefore, the proposed method has better performance in improving the localization accuracy.

In future work, we can consider whether the exact relation between the threshold t and CS theory can be derived by a rigorous theoretical analysis and formulation so that the optimal threshold can be obtained, which can enable the optimal performance of the sensing matrix in CS theory.

## Figures and Tables

**Figure 1 entropy-25-01025-f001:**
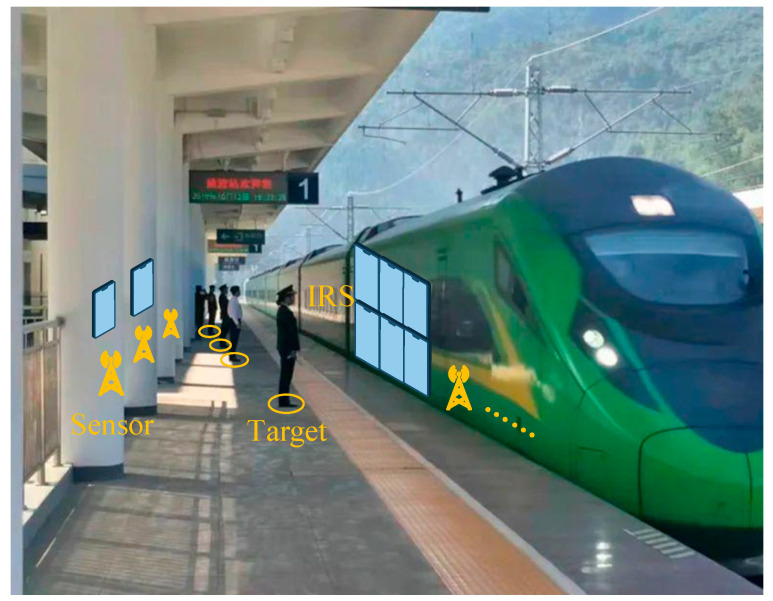
Photograph of the train entry station.

**Figure 2 entropy-25-01025-f002:**
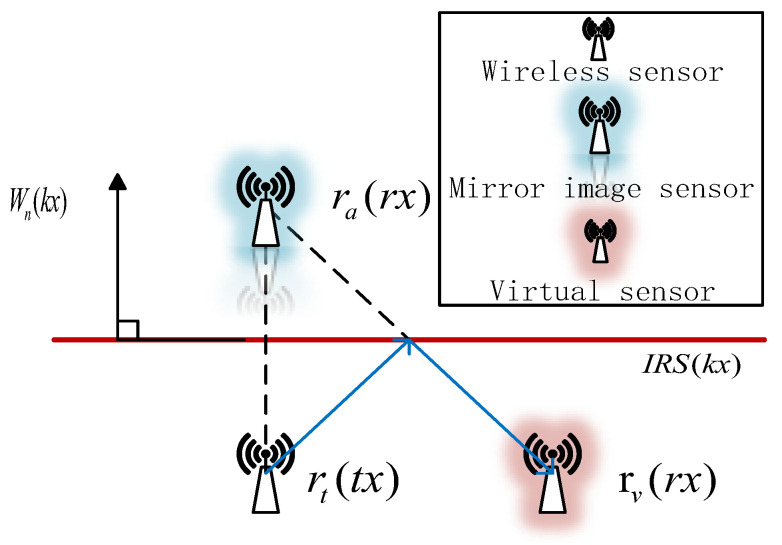
Schematic diagram of virtual sensor model.

**Figure 3 entropy-25-01025-f003:**
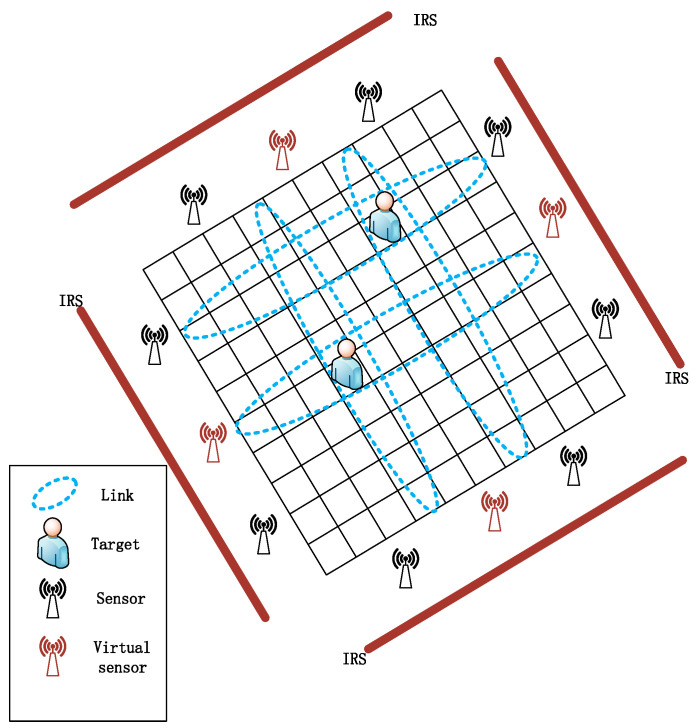
Device-free localization scenario.

**Figure 4 entropy-25-01025-f004:**
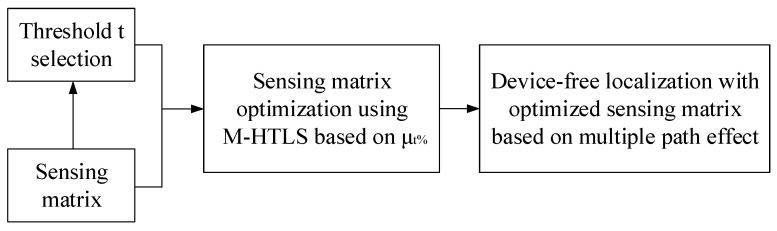
Flow chart of the optimization method.

**Figure 5 entropy-25-01025-f005:**
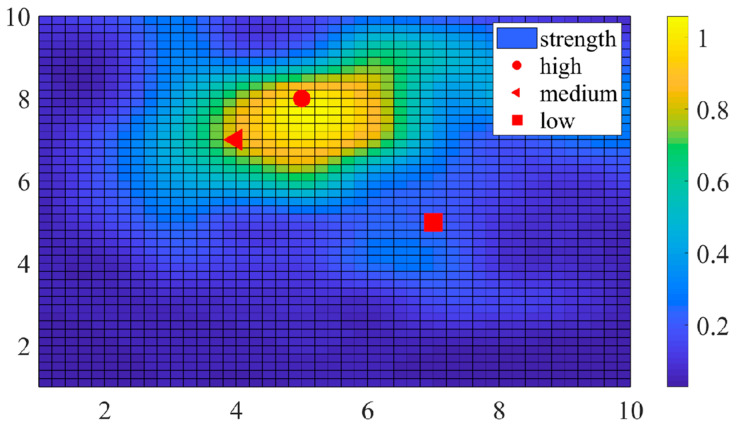
Similarity between the 58-th column and other columns in Φ.

**Figure 6 entropy-25-01025-f006:**
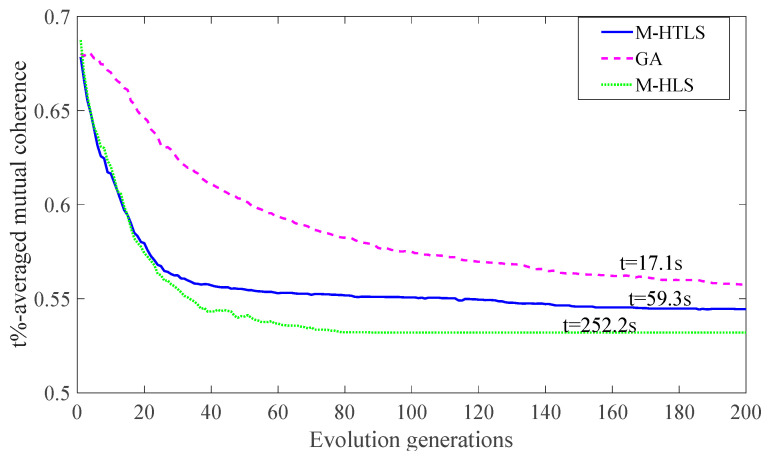
Performance comparison among M-HTLS, GA, and M-HLS.

**Figure 7 entropy-25-01025-f007:**
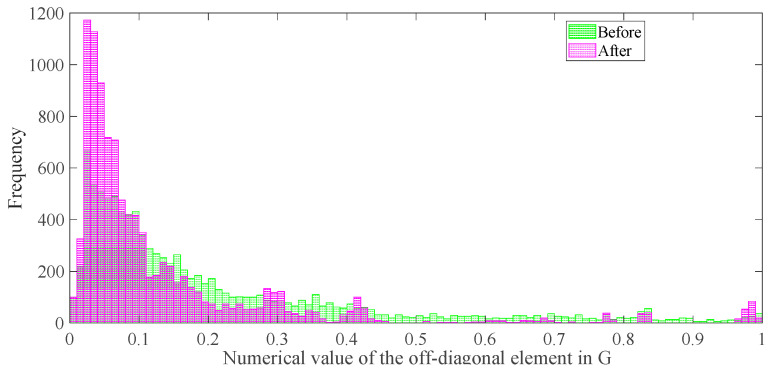
Comparison of the off-diagonal entries of the sensing matrix before and after M-HTLS optimization.

**Figure 8 entropy-25-01025-f008:**
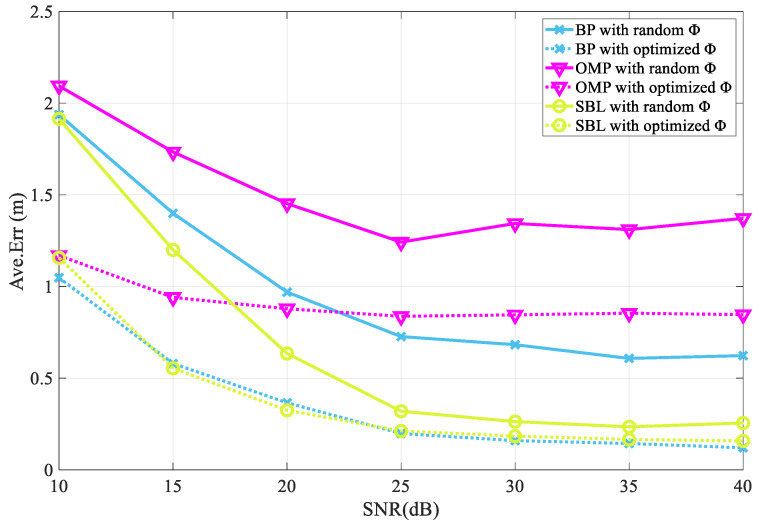
Optimization effect of M-HTLS on BP, OMP, and SBL sparse recovery methods under different SNRs.

**Figure 9 entropy-25-01025-f009:**
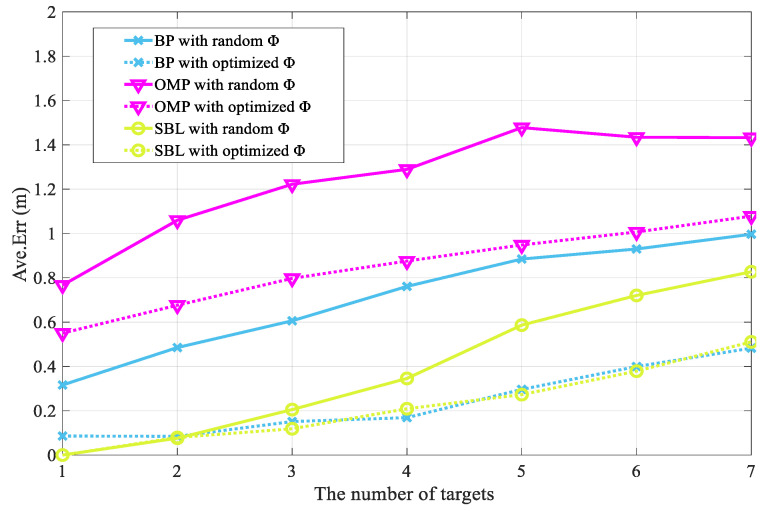
Optimization effect of M-HLTS on BP, OMP, and SBL sparse recovery methods with different number of targets.

**Figure 10 entropy-25-01025-f010:**
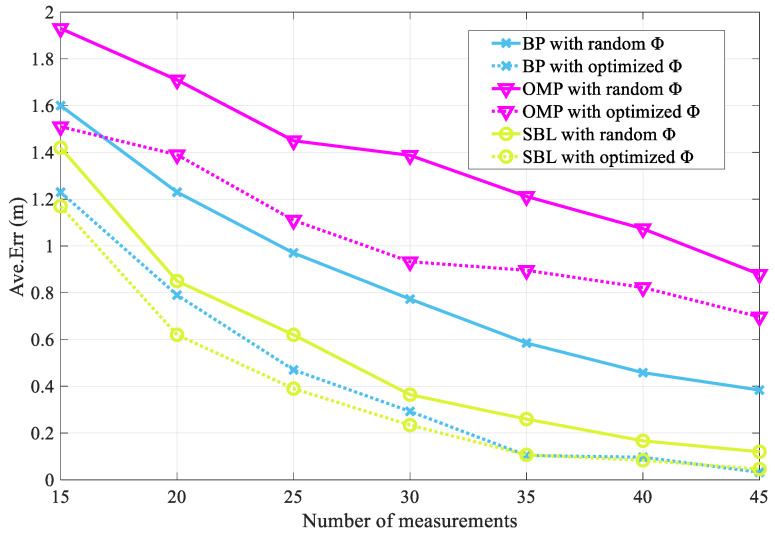
Optimization effect of M-HTLS on BP, OMP, and SBL sparse recovery methods with different number of sensing matrices.

## Data Availability

Data available on reasonable request.

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
