# Peer review of "Meta-Heuristic Device-Free Localization Algorithm under Multiple Path Effect"

_entropy, 2023, doi:10.3390/e25071025_

Round 1
Reviewer 1 Report
The authors present an interesting paper entitled "Meta-Heuristic Device-Free Localization Algorithm under Mul- 2 tiple Path Effect". Please consider the following concerns.
Major concerns:
- Please, clearly describe all parameters used in the equations.
- Make sure that figure legends provide a clear description of each figure. E.g., the legend of Figure 5 doesn't give any information to understand the chart.
- Section 4 is hard to follow. Please try to make it more reader-friendly
English concerns:
- Please avoid the use of acronyms in the abstract.
- Something is wrong with the references in the text.
- Please replace the characters in line 175 with English text.
Reviewer 2 Report
The paper addresses processing signals from different sources for localisation of targets using multi-path effects to generate additional - virtual - sensors. These sensors are part of so called "intelligent" surfaces, where the term „intelligent“ should be explained in detail.
The format of the paper must be completely reviewed and enhanced, e.g. the authors need to clearly indicate the references. Otherwise the paper is not readable. In addition, the equations are often not aligned with the text and mathematical symbols are changing the line spacing in the paragraph. So the layout should be carefully checked and corrected, e.g. line 491/493 and following. The spacing of lines is different in most of the paragraphs and should be the same for the complete paper.
Line 175: Please delete or translate the Chinese letters.
Line 178: What makes the surface „intelligent“? Please explain.
Line 187: What is algorithm 22?
Fig. 3: Why do the links have different colours?
The authors propose a sparse recovery algorithm using different matrices which need to be generated. Here it seems that we have a „measurement matrix“, a „sensing matrix“, a „perception matrix“ and even more. It is quite unclear how all these matrices are combined. An overview of how these matrices interact, would make it easier to understand the proposed method, e.g. by extending fig. 4?
Line 259: Are the gram matrices generated from the sensing matrices?
Line 290: Please check whether it makes sense to mention the performance twice.
Fig. 5: Please make clear which parameter is colour-coded in fig. 5. Is it Iij?
Line 354: Please explain what you mean by militaristic algorithms.
Line 441: „constantnd“?
Fig. 6: Why is the point x=54 highlighted in the diagram?
Fig. 7: Please check the format. The graphic is interfering with the describing text.
Line 478: Please explain the abbreviation ASDF.
Line 543: Does it make sense to provide the error down to mm range? The overall uncertainty of the localisation method is already beyond a mm.
Please check the extensive use of „scholars“.
Check the formulation of sentences e.g. like line 143.
Line 143: „the focus is focused“ Please check the meaning of the sentence.
Please check as well the grammar.
Reviewer 3 Report
This paper investigates the compressed sensing (CS) matrix optimization problem in a device-free localization scenario, based on multiple reflections. To solve this problem, one is modeled as a constrained combinatorial optimization problem, and a hybrid Meta-Heuristic Local Search (M-HTLS) algorithm is proposed. The properties of the measurement matrix are iteratively improved, while keeping it as a constrained binary sparse matrix. An optimization metric is used to evaluate the quality of each sensing matrix. It is shown that M-HTLS avoids generating additional transformation matrices, which do not lead to degradation of the signal-to-noise ratio compared to the conventional CS based localization methods. The simulation results show that the proposed method efficiently optimizes the sensing matrix and achieves fast and high-precision localization while conserving communication resources.
As some shortcoming and missing of the paper, it should be noted the following:
1. All references into text are shown incorrectly. For example:
(Line 29): “…rescue12.”
2. All abbreviations in the text must be deciphered at the site of first application. For example:
DFL (Line 59), CS (Line 85), RIP (Line 162), etc.
3. (Line 106): “…the mutual coherence of t% described in 14…” What is the t?
4. Abbreviations in the text must be the same: M-HTLS (Line 15) and M-THLS (Line 124).
5. (Line 175): Chinese characters???
6. (Line 183): What is the kx?
7. Figure 2: What is the rt?
8. Does the sensing matrix Ф (Lines 228, 229) coincide with the perception matrix Ф (Line 240)? If no, they should have different designations.
9. (Line 322): Do the coordinates [7, 5] represent d64?
10. Does the “militaristic algorithm” (Line 354) coincide with the “metaheuristic algorithm” (Line 356)?
11. Many used formulae and estimations are not obtained by the authors. In these cases, corresponding references must be placed into text. For example: estimation in Line 463.
12. There are incomplete references in References List and the MDPI rules for references are broken. All papers must contain doi.
Minor editing of English language required.
Round 2
Reviewer 2 Report
Please check whether your explanation of smart or intelligent surfaces fits in the paper.
Please check line 174 again - there are still Chinese signs for the reference.
Please check the title of fig.5 in line 327. Is it a header or a sub title? Please be consistent with the layout.
Line 477: Cannot be fig. 1, or?
The overall understanding is not well supported by the formatting of the text and the formulas. That makes it really hard to read and finally understand. Please check whether it could be enhanced.
English language is ok, moderate changes can be applied.
Reviewer 3 Report
The authors have introduced certain corrections into text, but several issues are remained.
1. (Line 105): It is remained the issue: “…the mutual coherence of t% described in [14]…” What is the t%?
The author’s reply should be introduced into text: “Here
refers to the concept of mutual coherence proposed by reference [14], which is explained as follows. The mutual coherence provides a measure of the worst similarity between the dictionary columns, a value that exposes the Dictionary’s vulnerability, as such two closely related columns may confuse any pursuit technique.”
2. (Line 174): Chinese characters???
3. Formula (1): What is the ra?
4. In References List, the MDPI rules for references are broken. All papers must contain doi.
Round 3
Reviewer 3 Report
The authors have performed necessary revision of the paper and it can be accepted in the present form.